# Study on Extraction, Physicochemical Properties, and Bacterio-Static Activity of Polysaccharides from *Phellinus linteus*

**DOI:** 10.3390/molecules28135102

**Published:** 2023-06-29

**Authors:** Nengbin Gao, Weijia Zhang, Dianjie Hu, Guo Lin, Jingxuan Wang, Feng Xue, Qian Wang, Hongfei Zhao, Xin Dou, Lihong Zhang

**Affiliations:** 1College of Food Science and Engineering, Changchun University, Changchun 130012, China; 220302128@mails.ccu.edu.cn (N.G.); zwj944858186@163.com (W.Z.); 220301080@mails.ccu.edu.cn (D.H.); 220301102@mails.ccu.edu.cn (G.L.); 18643605531@163.com (J.W.); wangqianya1997@163.com (Q.W.); 18809893865@163.com (H.Z.); douxin20222@163.com (X.D.); 2Jilin Province Changbai Forest Management Bureau, Baishan 134499, China; 19997151396@163.com

**Keywords:** *Phellinus linteus*, ultrasonic extraction, chemical analyses, anti-bacterial effect

## Abstract

We optimized an ultrasound-assisted extraction process of *Phellinus linteus* mycelium polysaccharides (PLPs) and studied their monosaccharide composition and bacteriostatic properties. Based on a single-factor experiment, a three-factor, three-level Box–Behnken design was used to optimize the ultrasound-assisted extraction process of PLP, using the yield of PLP as the index. The chemical composition and monosaccharide composition of PLP were determined by chemical analysis and HPLC analysis, respectively. Microscopic morphological analysis of the surface of PLP was performed via swept-surface electron microscopy. The bacteriostatic properties of PLP were determined using the spectrophotometric turbidimetric method. The results showed that the best extraction process of PLP with ultrasonic assistance achieved a result of 1:42 g/mL. In this method, the ultrasonic temperature was 60 °C, ultrasonic extraction was performed for 20 min, and the yield of PLP was 12.98%. The monosaccharide composition of PLP mainly contains glucose (Glc), mannose (Man), galactose (Gal), and glucuronic acid (GlcA). The intracellular polysaccharide of Phellinus igniarius Mycelia (PIP) is an irregular spherical accumulation, the surface is rough and not smooth, and the extracellular polysaccharide (PEP) is a crumbly accumulation. PIP has a stronger inhibitory ability for *S. aureus* and *E. coli* and a slightly weaker inhibitory effect for *B. subtilis*; the inhibitory effect of PEP on *S. aureus*, *E. coli*, and *B. subtilis* is slightly inferior to that of PIP.

## 1. Introduction

*Phellinus linteus* (PL) is a rare large fungus. PL is also known as mulberrychen, tree chicken, plum tree fungus, etc., and belongs to the fungus world, stretcher fungus, umbrella fungus, and rust Gekomycetes. *Phellinus linteus* genus is mainly produced in tropical America, Africa, and East Asia [1]. As one of many medicinal mushrooms, it is widely used in East Asia, especially in Korea, China, and Japan [2]. The description of the pharmacological activity of PL can be traced back to Shennong’s Herbal Classic of Materia Medica more than 2000 years ago. It was also recorded in the Compendium of Materia Medica by Li Shizhen of the Ming Dynasty that “Mature PL is available in yellow, white and golden yellow, all of which can be used for medicinal purposes”. It is used to treat bleeding and stop bleeding, gastrointestinal dysfunction, spleen deficiency, diarrhea related to female menstruation, etc. [3]. The Compendium of Materia Medica classifies PL as a vegetable and suggests that the fungus is hard in texture as Chen and Huang, and the fungus is soft as Chicken, E, or Curcuma. Thus, the yellow of PL refers to the color or texture.

Many of the plants, animals, and fungi distributed in nature contain specific natural active substances such as polysaccharides [4]. Polysaccharides are formed by the polymerization of different monosaccharides through complex chemical bonds [5]. Polysaccharides provide energy to living organisms and are indispensable substances. Polysaccharides are also involved in the composition of the body, and glycoproteins play an essential role in cells and are formed from polysaccharides and proteins [6]. Due to the growth habit of the fungus, it is possible to reduce the preparation time of fungal polysaccharides drastically. They can be used in industrial production and have good market prospects. In recent years, fungal polysaccharides have attracted great interest due to their biological activity and health effects [7]. 

Meanwhile, the main active component of PL is a polysaccharide that has a variety of biological activities. These include anti-oxidation [8], antitumor [9], immunomodulatory [10,11], anti-inflammatory [12], anti-virus [13], free radical scavenging ability [14], and antibacterial effects [15]. An increasing number of structurally diverse polysaccharides have been obtained from *Phellinus linteus* cotyledons, *Phellinus linteus* mycelium, and fermentation broth by a large number of extraction and separation methods, such as hot water extraction, ultrasonic extraction, enzymatic extraction, and a large number of purification methods such as ethanol precipitation and packed column chromatography [16].

## 2. Results and Discussion

### 2.1. Glucose Labeling Curve Determination

The standard curve was made using glucose (Glc) as the standard, and the regression equation was:Y = 0.0085X + 0.0049 (R^2^ = 0.9993).

### 2.2. Effect of Ultrasonic Time on the Yield of PLP

As shown in Figure 1, at 200 W, 1:20 g/mL, and 50 °C extraction, the yield of PLP was increased and then decreased, and the highest yield was obtained at 20 min. When the ultrasonic time was higher than 20 min, the yield of PLP gradually decreased. The reason for this situation was that, with the combined effect of ultrasound and heating, the cell wall of the sample was gradually destroyed, and PLP was dissolved continuously. Meanwhile, the cavitation and mechanical effects of ultrasonic waves could also promote the release and diffusion of PLP in water, and the polysaccharide dissolution rate would increase accordingly, thus showing a trend towards a higher extraction rate. However, on the other hand, the cavitation effect of ultrasonic waves can also promote the oxidation and degradation of PLP and the dissolution of impurities, which decreases the polysaccharide extraction rate. Therefore, we chose 10, 20 and 30 min as the ultrasonic time optimization levels for the response surface optimization experiment.

### 2.3. Effect of Solid–Liquid Ratio on the Yield of PLP

As shown in Figure 2, The extraction was carried out at an ultrasonic power of 200 W and an ultrasonic temperature of 50 °C for 20 min, and the polysaccharide yield was highest at a solid–liquid ratio of 1:40 g/mL with the increase in the solid–liquid ratio, and slightly decreased after exceeding 1:40 g/mL. The analysis of the reason for this situation is that the increase in solid–liquid ratio increases the solvent concentration difference between inside and outside of PL cells, and the larger concentration difference can increase the polysaccharide diffusion coefficient, thus promoting faster and more dissolution of PLP. In addition, more solvents increase the range of action of ultrasound, which also contributes to the increase in polysaccharide extraction rate. However, the extraction rate of polysaccharides started to decrease after the solid–liquid ratio exceeded 1:40 g/mL, which was due to the decrease in ultrasonic energy per unit volume by too much solvent and the decrease in ultrasonic force, which led to the decrease in polysaccharide extraction rate. Therefore, we chose 1:30, 1:40 and 1:50 g/mL as the solid–liquid ratio optimization levels for the response surface optimization experiment.

### 2.4. Effect of Ultrasonic Temperature on the Yield of PLP

As shown in Figure 3, the yield of PLP increased and then decreased with the change in sonication temperature under the conditions of sonication power of 200 W, a solid–liquid ratio of 1:40 g/mL and sonication time of 20 min, and the highest yield of PLP was obtained at 60 °C. It has been reported in the literature that high temperatures can disrupt the plant cell wall and reduce the viscosity of the solution, promoting better polysaccharide solubilization from the cells and thus improving the polysaccharide extraction rate. However, too high extraction temperature will also lead to the degradation of high molecular weight polysaccharides [17]. As shown in Figure 3, the extraction rate decreased slightly when the extraction temperature exceeded 60 °C, which was attributed to the high temperature completely rupturing the PL cell wall. The polysaccharide solubilization rate had reached essential stability. Further, an increase in temperature would have little effect on PLP solubilization but would lead to polysaccharide degradation and activity destruction. Therefore, temperatures of 50, 60, and 70 °C were selected for the next response surface experimental optimization step.

### 2.5. Response Surface Test of Extraction PLP by Ultrasonic-Assisted Extraction

#### 2.5.1. Response Surface Test Protocol and Results

A mathematical model was developed for the yield of PLP using Design-Expert 8.0.6 software Box–Behnken design factor levels to optimize the process parameters for ultrasound-assisted extraction of PLP.

With ultrasonic time (A), solid–liquid ratio (B), and ultrasonic temperature (C) as independent variables and PLP extraction rate as response values, the data of each factor were fitted by regression according to the response surface test analysis scheme in Table 1 to establish a quadratic polynomial equation for the yield of PLP:Yield% = 12.96 + 0.074A + 0.46B + 0.17C − 0.095AB − 0.28AC − 0.24BC − 1.53A^2^ − 1.08B^2^ − 1.02C^2^

#### 2.5.2. Analysis of Variance

As shown in Table 2, the model F value was 392.81, and the *p* value was less than 0.01, indicating that the second model test was reliable and could accurately and appropriately reflect the changing relationship between each factor and the response value. The model determination coefficient R^2^ = 0.9980, and the misfit term is insignificant, indicating that the model is stable and reliable and can be used to guide practical application analysis. B, C, AC, and BC had a highly significant effect (*p* < 0.01), A had a significant effect (*p* < 0.05), while AB had no significant effect (*p* > 0.05) on the extraction of PLP. Comparing the F-values of three factors A, B, and C, the effect of each factor on the yield of PLP was in the following order: B > C > A, i.e., solid–liquid ratio > ultrasonic temperature > ultrasonic time.

#### 2.5.3. Results of Interactions between Factors

The steeper the response surface plot is, the more significant the influence of the factor is, and conversely, the flatter it is, the less significant the influence. The contour map can be judged by the shape of the graph to determine the degree of interaction; if the graph is elliptical, the more pronounced the interaction phenomenon, and if the graph is circular, the interaction phenomenon is not apparent. From Figure 4, it can be seen that the order of the curvature of the response surface plot surface is AC > BC > AB, and both the ultrasonic time and ultrasonic temperature have a highly significant effect on the yield of PLP, which is consistent with the ANOVA results of the regression equation.

#### 2.5.4. Validation Experiments

The regression equation was solved by Design-Expert 8.0.6 software, and the optimal extraction process conditions of PLP were obtained: the model predicted the yield of PLP to be 13.01% with a solid–liquid ratio of 1:42.05 g/mL, an ultrasonic temperature of 60.59 °C and ultrasonic time of 20.13 min. Considering the feasibility of operation in the actual application, the ultrasonic time was adjusted to 20 min, material–liquid ratio 1:42 g/mL and ultrasonic temperature 60 °C. The average yield of PLP was 12.98% after three parallel experiments to verify the test results. The measured results are close to the predicted results, the surface model fits well, and the parameters have high confidence.

### 2.6. Physicochemical Properties of PLP

#### 2.6.1. Results of Total Sugar Content Measurement

The glucose standard curve of the regression equation is Y = 0.0085X + 0.0049 (R^2^ = 0.9993). The linear regression equation with a correlation coefficient of R^2^ = 0.9993 indicates that the phenol–sulfuric acid method can accurately determine the total sugar content of samples in the range of 10–60 ug/mL. The absorbance of the PIP and PEP measured at 490 nm were brought into the glucose standard curve. It was calculated that the total sugar content in the PIP was 91.07%, and the total sugar content in the PEP of liquid fermentation was 81.54%. PIP has a higher total sugar content than PEP.

#### 2.6.2. Results of Polysaccharide UV Full Wavelength Scan Analysis

The two polysaccharide solutions of 1 mg/mL of PIP and PEP were scanned at 200–800 nm at medium speed, and the results are shown in Figure 5. Generally, proteins have absorption peaks at 275–280 nm, and nucleosides have strong absorption peaks at 240–290 nm. The polysaccharide absorption peaks of the two polysaccharides were found at 206 nm, but the rest of the section did not show any absorption peaks, indicating that the protein content of the two polysaccharides was minimal. Therefore, it is not subject to protein and nucleoside removal and is considered a pure polysaccharide.

#### 2.6.3. Protein Content Measurement Results

The protein standard curve of the regression equation is
Y = 0.0105X + 0.0443 (R^2^ = 0.9994).

The PIP and PEP absorbance at 595 nm was brought into the linear regression equation of the standard curve for protein content determination. It was calculated that the protein content in the PIP was 1.11%, and the protein content in the PEP was 0.86%. PIP and PEP have less protein and can be treated as pure polysaccharides.

#### 2.6.4. Ash Content Measurement Results

The PIP and PEP were accurately weighed at 30 mg in a crucible and weighed after sintering at 550 °C for 8 h. The ash content of PIP was calculated to be 7.59%, and the ash content of PEP was 8.96%. The ash content of PIP was lower than that of PEP.

#### 2.6.5. Results of Monosaccharide Composition Determination

As shown in Figure 6, the monosaccharide composition of the PIP and PEP was determined by HPLC. We speculate that the two polysaccharides are heteropolysaccharides made up of multiple monosaccharides.

As shown in Table 3, We speculate that the PIP was mainly composed of glucose, mannose and galactose. The three monosaccharides’ total content accounted for 97.2% of the total sugar content. It is consistent with the experimental results on the monosaccharide composition of Jiang Peng [18]. The PIP consisted of six monosaccharides with a molar ratio of Glc:Man:Gal:Fuc:GlcA:Xyl = 87.6:5.4:4.2:1.4:0.8:0.6. The PEP was mainly composed of glucose, mannose, galactose and arabinose, and the four monosaccharides together accounted for 99.1% of the total sugar content. The PEP may have been composed of only five monosaccharides with a molar ratio of Glc:Man:Gal:Ara:GlcA = 93.6:1.9:1.8:1.8:0.8. The monosaccharide composition of both polysaccharides had the highest percentage of glucose, and arabinose was the unique monosaccharide component of the PEP. Kozarski et al. also used the hot water extraction method to obtain PLP [19]. The results showed that PLP consisted of Fuc:Rha:Ara:GlcN:Gal:Glc:Xyl = 1.4:0.5:0.9:1.6:4.7:84.8:6.0. Slightly different from PIP and PEP, the reason for this difference may be the effect of different extraction methods on the monosaccharide composition and possibly the difference between mycelium and the fruiting bodies.

#### 2.6.6. Scanning Electron Microscope Results Analysis

We used scanning electron microscopy to analyse the PIP and PEP microscopic morphology of *Phellinus linteus* mycelium. It can be seen in Figure 7 that the two polysaccharides showed small microstructures under 100× electron microscopy, and the morphology of the two polysaccharides was different, with most of the PIP in an irregular spherical shape and the PEP in small crumbly accumulation.

In Figure 8, it can be seen that in the two polysaccharides under 500× electron microscopy, the PIP was irregularly spherical, with a rough and uneven surface and a small number of small spherical particles attached to its surface, and the PEP was similar to the surface of mycelium polysaccharide. Both were uneven, with a small number of bumps, and the overall shape was irregular.

Figure 9 shows the two polysaccharides in 1000× electron microscopy. The PIP is irregularly geometric, with irregular bumps attached to its surface and an uneven and rough surface, and the PEP is an irregular three-dimensional structure with apparent bumps. The surfaces of the two polysaccharides were not smooth and uniform, and all of them were perforated and folded structures, which might be caused by specific repulsive forces between the molecules of the 2 PLP and the small intermolecular attraction.

### 2.7. Results of the Analysis of the Antibacterial Properties of PLP

As can be seen in Figure 10, the PIP gradually decreased the inhibition rate of *S. aureus* over time, and the PEP was extended with the shaking culture time; the trend was the same as the PIP, and the inhibition rate gradually decreased. PIP showed the strongest inhibition of *S. aureus*, with 80% inhibition at 1.0 h. The inhibition rate of the PIP against *E. coli* was the highest at 1.5 h, with 75.47%. The inhibition of *E. coli* by PEP gradually decreased with time. The two polysaccharides were slightly less effective in inhibiting *B. subtilis*, and the inhibition rate gradually decreased with time. The PIP had the best effect on its inhibition, with the highest inhibition rate of 67.74%. The inhibitory effect of the PIP on *B. subtilis* was more potent than that of the PEP.

## 3. Materials and Methods

### 3.1. Chemicals and Reagents

Monosaccharide standards, including Arabinose (Ara), Fucose (Fuc), Galactose (Gal), Galacturonic acid (GalA), Glucose (Glc), Glucuronic acid (GlcA), Mannose (Man), Rhamnose (Rha), and Xylose (Xyl), of chromatographic purity were purchased from Shanghai Yuanye Bio-Technology Co. (Shanghai, China). Anhydrous ethanol, glucose, phenol, sulfuric acid, and 1-Phenyl-3-methyl-5-pyrazolone (PMP) of analytical purity were purchased from Sinopharm Chemical Reagent Co. (Shanghai, China). Acetonitrile of chromatographic purity was purchased from Tianjin Chemical Reagent Co. (Tianjin, China). coomassie brilliant blue G250 of analytical purity purchased from Beijing Dingguo Bio (Beijing, China). Pancreatic peptone, yeast extract powder, beef extract, and biochemical reagent were purchased from Beijing Auboxing Biotechnology Co. (Beijing, China). Bovine serum albumin of biotechnology grade was purchased from Shanghai Maclean Biochemical Technology Co. (Shanghai, China). Pure water was used during the experiments.

### 3.2. Determination of the Glucose Standard Curves

The glucose standard was precisely measured at 100.0 mg, dried to constant weight at 105 °C, placed in a 100 mL volumetric flask, added to water to fix the glucose standard to the scale, shaken well, and configured into 1.0 mg/mL glucose solute. Then, 10 mL of it was moved to a 100 mL volumetric flask, to which water was added to fix the volume to 100 mL, and it was configured into a 0.1 mg/mL glucose solution. Then, 0.0, 0.2, 0.4, 0.6, 0.8, 1.0, and 1.2 mL of the solution was added to a stoppered test tube, water was added to make up 2.0 mL, and 1.0 mL of 5% phenol solution was added. The solution was shaken well, and then 5.0 mL of sulfuric acid was quickly added and shaken well. This was left to stand for 30 min, and the absorbance was measured at 490 nm [20].

### 3.3. The Extraction Process of PLP

PL fungus was inoculated into liquid medium (MgSO_4_ 0.5 g/L, KH_2_PO_4_ 0.5 g/L, NaCl 1.0 g/L, sucrose: beef extract 40:10 (g/g) with water to 100 mL) at 26 °C, 150 r/min incubator for 9 days.

The PL fermentation broth was centrifuged to obtain the supernatant, and the concentrate was obtained by rotary evaporation. Four times the volume of ethanol was added, and this was placed at 4 °C overnight, centrifuged and precipitated by freeze-drying to obtain PEP powder.

The appropriate amount of PL powder was weighed and extracted by ultrasonic-assisted hot water extraction. The supernatant was rotary-evaporated to obtain the concentrate, and then 4 times its volume of ethanol was added. This was placed at 4 °C overnight, centrifuged and precipitated by freeze-drying to obtain PIP powder.

The appropriate amount of *Phellinus linteus* mycelia powder was weighed, which was extracted by ultrasonic-assisted hot water extraction to obtain a supernatant. The excess water in the polysaccharide solution was removed by rotary evaporation, concentrated to one-fifth of the original volume, and 4 times the volume of ethanol was added This was placed at 4 °C overnight and centrifuged at 3500 rpm/min for 30 min, and then the supernatant was discarded, leaving the precipitate, which was evaporated at 60 °C to remove the residual ethanol on its surface, and freeze-dried to obtain polysaccharide powder [21].

### 3.4. Single-Factor Test on the Extraction of PLP

When the ultrasonic power was fixed at 200 W, three factors of ultrasonic time, solid–liquid ratio, and ultrasonic temperature were selected for the single-factor test. The effects of ultrasonic time (10, 20, 30, 40 and 50 min), material–liquid ratio (1:10, 1:20, 1:30, 1:40 and 1:50 g/mL) and ultrasonic temperature (40, 50, 60, 70 and 80 °C) on the extraction of PLP were studied under the same process conditions, and the same amount of raw materials were added [22,23].

### 3.5. Response Surface Test on the Extraction of PLP

The Box–Behnken experimental design was applied to Design-Expert V8.0.6 software to determine the suitable process parameters for ultrasound-assisted extraction of PLP by response surface test analysis with PLP yield as the response value [24,25]. Experimental factors and level design are shown in Table 4.

### 3.6. Physical and Chemical Property Analysis of PLP

#### 3.6.1. Determination of Total Sugar Content

The samples were prepared as 0.1 mg/mL polysaccharide solution, and 1 mL was drawn into the test tubes, respectively, according to the glucose standard curve preparation method. After the sample was measured, the average absorbance of the sample was calculated and substituted into the standard curve to calculate the total sugar content in the sample [26,27].

#### 3.6.2. PLP UV Full Wavelength Scan Analysis

The principle of full wavelength scanning is to measure the sample in one wavelength range and reflect the absorbance of the sample at different wavelengths. The PIP and PEP were configured into 1 mg/mL polysaccharide solution. The scanning mode was selected as ABS with distilled water as a blank control, the scanning interval was 1 nm, the scanning speed was medium, and the sample to be measured was inserted. The full wavelength spectrogram of the sample could be obtained by its UV full wavelength scan at 200–800 nm [28].

#### 3.6.3. Determination of Protein Content

Bovine serum albumin was prepared as a 5 mg/mL solution and diluted 100 times. Then, quantities of 0.0, 0.1, 0.2, 0.3, 0.4, 0.5, and 0.6 mL of the solution were added to a stoppered test tube, to which 1.0 mL of water was added to each; then, 2.5 mL of 0.1 mg/mL of coomassie brilliant blue G250 was added to the test tube, and this was shaken well before being left to stand for 5 min at room temperature. The absorbance of the sample was measured at 595 nm, and the standard curve was plotted using the mass of bovine serum albumin it contained (μg) as the horizontal coordinate and the corresponding absorbance as the vertical coordinate [29].

The samples were configured into 1 mg/mL polysaccharide solution, respectively, and 1 mL of each sample was sampled in the test tube. The operation procedure was referred to as the protein standard curve configuration method. After measuring the samples, the average absorbance was calculated and substituted into the standard curve to calculate the protein content in the samples.

#### 3.6.4. Determination of Ash Content

An empty crucible was cauterized in a muffle furnace at 550 °C for 12 h. After cooling, the crucible was removed, and its mass was accurately weighed. Then, 30 mg of the sample was put into the crucible to be measured, and the crucible was seared at 550 °C for 8 h. After cooling, the crucible was removed and the total mass weighed. The ash content of the sample was calculated according to Equation (1), where: *m*_1_ is the mass of the empty crucible after sintering (g); *m*_2_ is the total mass of the crucible and ash (g); and *m* is the mass of the sample to be measured (g).
(1)$$\mathrm{Ash}\;\mathrm{content}/\%=\frac{m_{\;2\;}-{\;m}_{1}}{m}\;\times \;100\%\;$$

#### 3.6.5. Determination of Monosaccharide Composition

An amount of 2 mg of the sample was added to the acid hydrolysis vial. To this, 1 mL of methanolic solution of hydrochloric acid was added. This was charged with N_2_ and reacted for 16 h at 80 °C in a constant temperature metal bath. After the reaction, the methanol of the hydrochloric acid in the sample was blown dry with a nitrogen-blowing instrument. Then, 1 mL of 2 M trifluoroacetic acid (TFA) was added to the reaction at 120 °C for 1 h. After the reaction, it was blown dry again. Next, monosaccharide derivatization was performed by adding 0.5 mL of 0.3 M NaOH to the acid hydrolysis vial to dissolve the dried monosaccharide sample inside completely. Then, 0.5 mL of 0.5 M PMP-methanol was added, at which time PMP and NaOH diffused rapidly, and this was mixed. The insoluble material at the bottom was blown with a pipette to disperse it evenly, and then 0.2 mL of the mixture was aspirated in an EP tube. The sample was then put into a water bath at 70 °C for 30 min. Samples were taken after the reaction and extracted after adding 0.1 mL of 0.3 M HCl. Then, 0.7 mL of CH_2_Cl_2_ was added, and the EP tube was shaken for 90 s until complete mixing and centrifugation to completely separate the organic phase from the aqueous phase. The lower organic phase was pumped clean with a flat-tipped syringe; this was repeated twice. The remaining aqueous phase was pumped up with a 1 mL syringe and then filtered through an organic filter membrane of 0.22 μm, and the sample was subsequently placed in HPLC for detection [30,31,32].

HPLC analysis method: Shimadzu HPLC system (LC-20ATvp pump and SPD-20AVD UV detector), COSMOSIL 5C18-PAQ column (4.61 × 250 mm), mobile phase PBS (0.1 M, pH 7.0)-acetonitrile 80.8:19.2 (*v*/*v*) at a flow rate of 1.0 mL/min, with an injection volume of 20 μL and detection at 245 nm.

#### 3.6.6. Scanning Electron Microscopy Analysis of PLP

An appropriate amount of polysaccharide sample was taken, placed in an ion sputterer, coated with a layer of conductive gold powder on the surface, and placed under an electron scanning microscope for observation. The sample interference and systematic errors were eliminated to observe the polysaccharide surface morphology. The observation magnification was chosen to be 100×, 500× and 1000×, the electron gun acceleration voltage was 20 kV, and the resolutions were 100 μm, 50 μm and 10 μm, respectively [33,34].

### 3.7. Analysis of the Antibacterial Properties of PLP

A total of 10.0 g of pancreatic peptone, 5.0 g of yeast extract powder, and 10.0 g of NaCl were accurately weighed. To this, distilled water was added to make up 1 L, which was sterilized, and liquid LB medium was prepared. Under aseptic conditions, cryopreserved *E. coli*, *S. aureus* and *B. subtilis* were added to the LB medium and activated at 200 rpm, 37 °C for 12 h. Then, 0.5 mL of each activated bacterial solution was taken and diluted with sterile saline to 10^7^ CFU/mL~10^8^ CFU/mL. The bacterial suspension was then put in a sterile EP tube and shaken evenly.

The sterile centrifuge tube was boiled at 100 °C for 20 min and sterilized by UV irradiation for 30 min, and 0.5 mL of each polysaccharide solution was taken. Equal amounts of *S. aureus*, *E. coli*, and *B. subtilis* were added, shaken at 37 °C, 200 r/min and incubated at 0, 1.0, 1.5, 2.0, 2.5, 3.0, 3.5, 4.0 h, respectively. The samples were taken at 0, 1.0, 1.5, 2.0, 2.5, 3.0, 3.5, 4.0 h, the OD value was measured at 600 nm, and three parallel groups were made in each group. A line graph was plotted with time as the horizontal coordinate and inhibition rate as the vertical coordinate for analysis [35,36].
(2)$$\mathrm{Bacteria}\;\mathrm{inhibition}\;\mathrm{rate}/\%=\frac{{\mathrm{OD}}_{600\;\mathrm{nm}\;\mathrm{Control}\;}-{\;\mathrm{OD}}_{600\;\mathrm{nm}\;\mathrm{Experimental}}}{{\mathrm{OD}}_{600\;\mathrm{nm}\;\mathrm{Control}}}\;\times \;100\%\;$$

## 4. Conclusions

In the present study, an optimized ultrasonic-assisted extraction method was designed to extract PLP. Single-factor tests and response surface methodology were used to optimize the extraction process. The highest extraction rate of PLP was 12.98% under the conditions of a solid–liquid ratio of 1:42 g/mL, ultrasonic temperature of 60 °C and 20 min of ultrasonic time. Compared with the traditional hot water extraction method, the ultrasonic-assisted extraction method significantly reduced the extraction temperature, shortened the extraction time and improved the extraction rate [37]. The phenol sulfuric acid method determined the total sugar content of *Phellinus linteus* mycelium. The sugar content of PIP was 91.07%, and that of PEP was 81.54%, PIP has a higher sugar content than PEP. The coomassie brilliant blue method determined the protein content of PLP; the protein content of PIP was 1.11%, and that of PEP was 0.86%. The PIP ash content was 7.59%, and the PEP ash content was 8.96%. PLP contains fewer impurities and is a relatively pure heteropolysaccharide. HPLC was used to analyze the monosaccharide composition of PLP. PLP mainly comprises glucose, mannose and galactose, with low contents of other monosaccharides. Scanning electron microscopy analysis of the microscopic morphology of PLP revealed that the PIP showed irregular spherical accumulation with a rough and uneven surface, and the PEP showed crumbly accumulation with a similar uneven surface as the PIP. The inhibition rate of PLP against *S. aureus*, *E. coli* and *B. subtilis* indicated that both PIP and PEP had good inhibition effects. This study proposed a reliable polysaccharide extraction process, and the resulting PLP has potential biological activities, which provide a theoretical basis for guiding clinical applications.

## Figures and Tables

**Figure 1 molecules-28-05102-f001:**
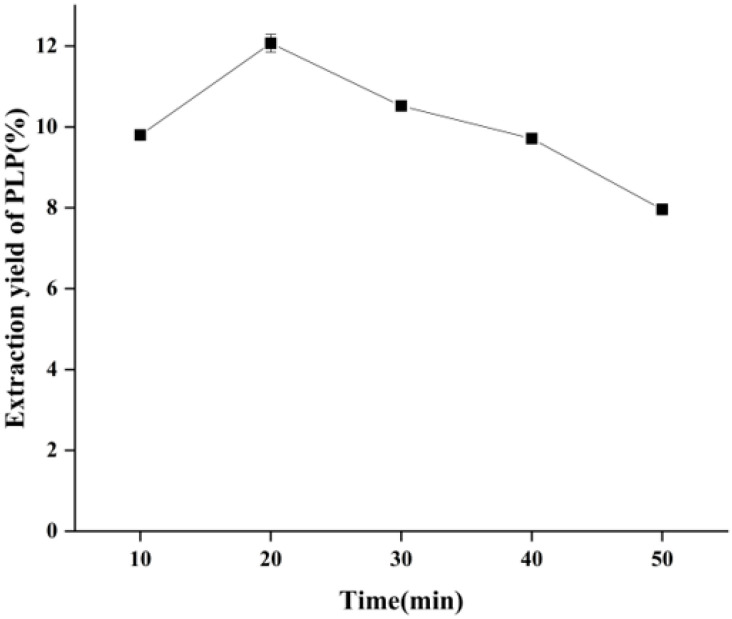
Effect of ultrasonic time on the yield of PLP.

**Figure 2 molecules-28-05102-f002:**
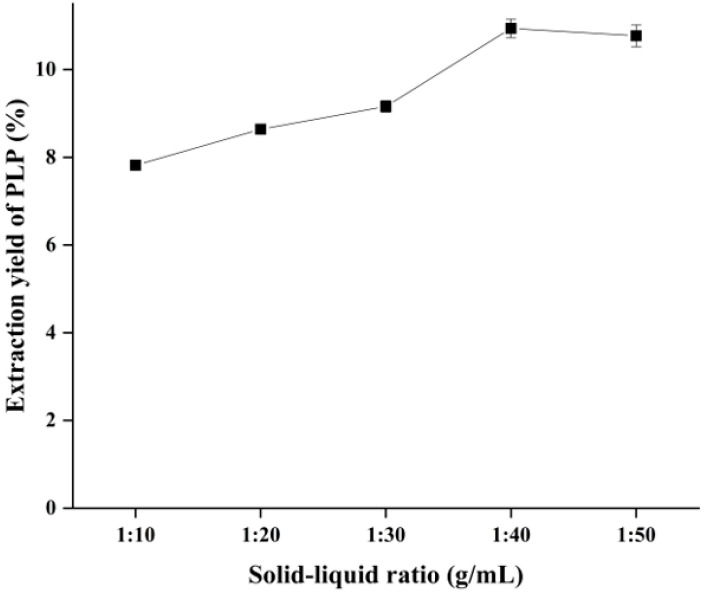
Effect of solid–liquid ratio on the yield of PLP.

**Figure 3 molecules-28-05102-f003:**
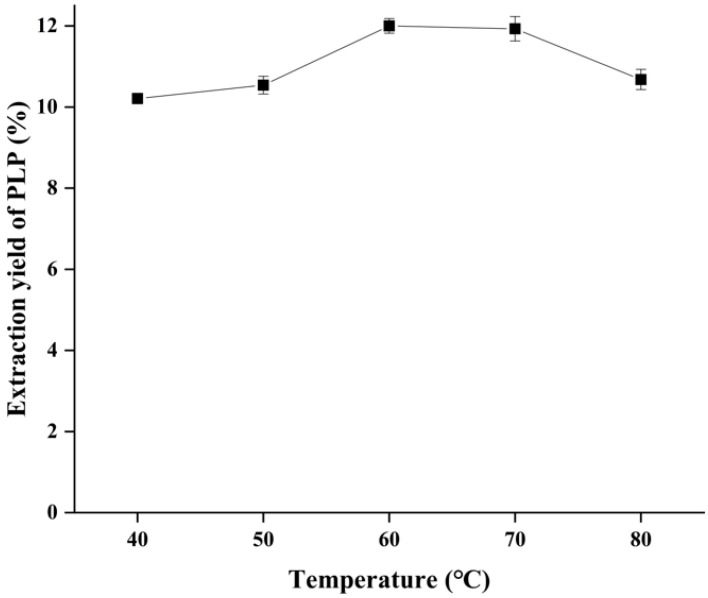
Effect of ultrasonic temperature on the yield of PLP.

**Figure 4 molecules-28-05102-f004:**
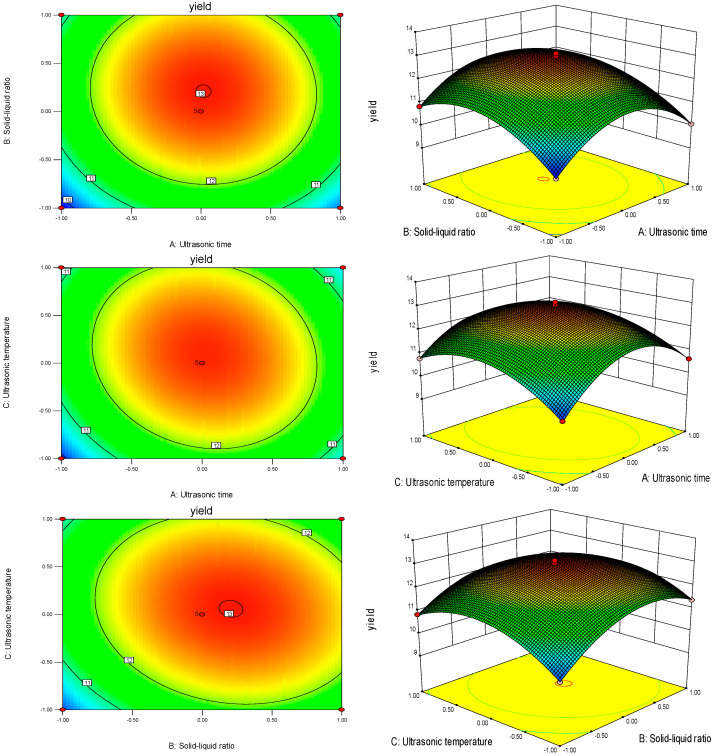
Effect of the interaction of factors on the yield of PLP.

**Figure 5 molecules-28-05102-f005:**
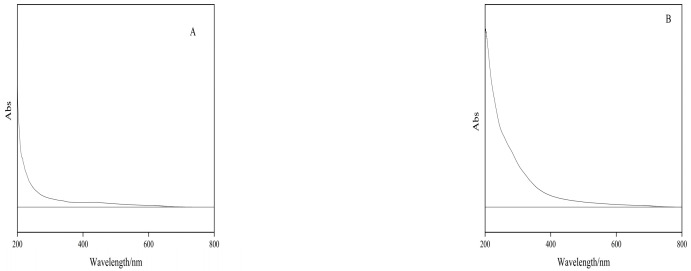
UV full wavelength scanning analysis of PIP and PEP. (**A**) PIP. (**B**) PEP.

**Figure 6 molecules-28-05102-f006:**
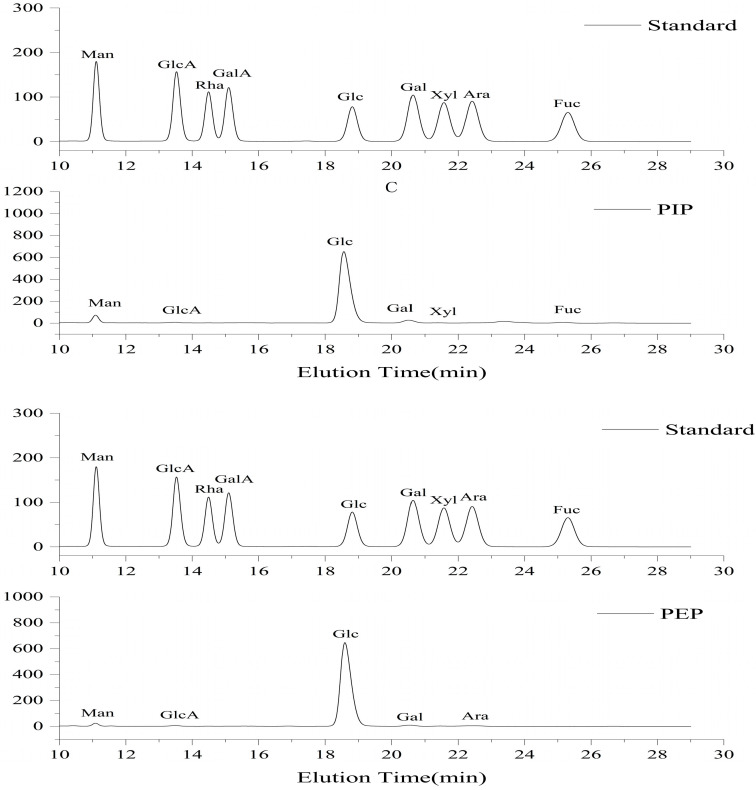
Analysis of monosaccharide composition of standard monosaccharides with PIP and PEP.

**Figure 7 molecules-28-05102-f007:**
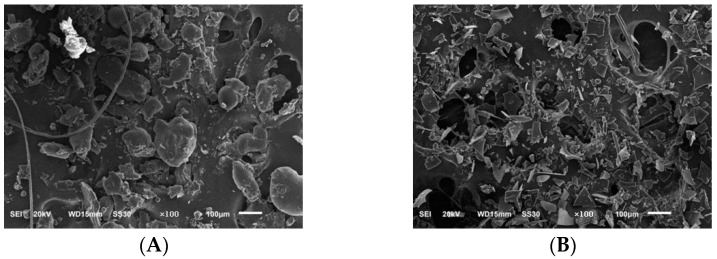
Scanning electron micrographs of PIP and PEP at ×100 magnification. (**A**) PIP. (**B**) PEP.

**Figure 8 molecules-28-05102-f008:**
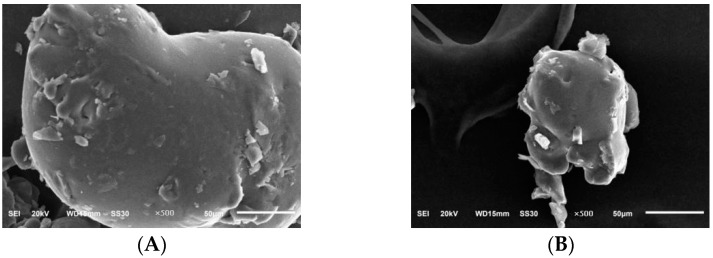
Scanning electron micrographs of PIP and PEP at ×500 magnification. (**A**) PIP. (**B**) PEP.

**Figure 9 molecules-28-05102-f009:**
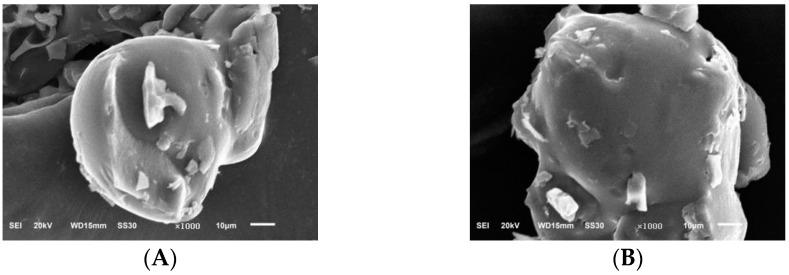
Scanning electron micrographs of PIP and PEP at ×1000 magnification. (**A**) PIP. (**B**) PEP.

**Figure 10 molecules-28-05102-f010:**
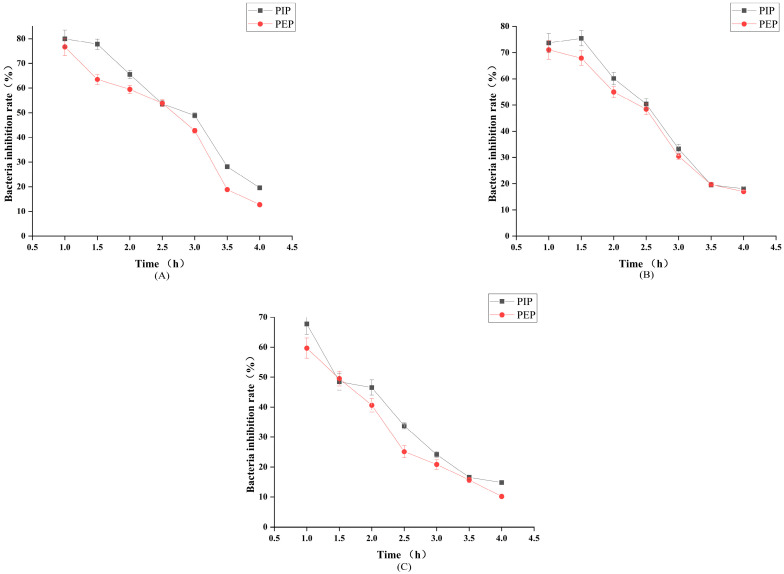
Antibacterial effect of PLP. (**A**) *S. aureus*. (**B**) *E. coli*. (**C**) *B. subtilis*.

**Table 1 molecules-28-05102-t001:** Response surface design model and test values.

Run	Independent Variables			Response
	A/(min)	B/(g/mL)	C/(°C)	Yield/(%)
1	1	1	0	10.79
2	0	0	0	12.82
3	0	0	0	12.99
4	0	1	1	11.27
5	0	−1	1	10.84
6	−1	1	0	10.83
7	0	−1	−1	9.98
8	0	0	0	12.9
9	−1	0	−1	9.9
10	0	0	0	13.1
11	0	0	0	12.97
12	1	0	1	10.35
13	−1	−1	0	9.72
14	1	0	−1	10.61
15	−1	0	1	10.77
16	1	−1	0	10.06
17	0	1	−1	11.36

**Table 2 molecules-28-05102-t002:** Variance analysis of regression model of response surface test.

Source	Sum of Squares	Df	Mean Square	F Value	*p*-Value Prob > F	Significance
Model	23.76	9	2.64	392.81	<0.0001	**
A	0.044	1	0.044	6.47	0.0384	*
B	1.67	1	1.67	247.79	<0.0001	**
C	0.24	1	0.24	35.42	0.0006	**
AB	0.036	1	0.036	5.37	0.0536	
AC	0.32	1	0.32	47.50	0.0002	**
BC	0.23	1	0.23	33.57	0.0007	**
A^2^	9.86	1	9.86	1467.53	<0.0001	**
B^2^	4.87	1	4.87	724.67	<0.0001	**
C^2^	4.36	1	4.36	649.26	<0.0001	**
Residual	0.047	7	0.007			
Lack of fit	0.003	3	0.001	0.10	0.9550	
Pure Error	0.044	4	0.011			
Cor Total	23.81	16				
R^2^	0.9980					

Note: ** is *p* < 0.01, the difference is highly significant; * is *p* < 0.05, the difference is significant.

**Table 3 molecules-28-05102-t003:** Monosaccharide composition of PLP (mol%).

Samples	Man	Clc-A	Glc	Gal	Xyl	Ara	Fuc
EPS	1.9	0.8	93.6	1.8	--	1.8	--
IPS	5.4	0.8	87.6	4.2	0.6	--	1.4

**Table 4 molecules-28-05102-t004:** Design factors and level of response surface test.

Independent Variable	Coded Levels		
	−1	0	1
Ultrasonic time (A) (min)	10	20	30
Solid–liquid ratio (B) (g/mL)	1:30	1:40	1:50
Ultrasonic temperature (C) (°C)	50	60	70

## Data Availability

The data used to support the findings of this study have not been made available.

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
