# Peer review of "Study on Extraction, Physicochemical Properties, and Bacterio-Static Activity of Polysaccharides from Phellinus linteus"

_molecules, 2023, doi:10.3390/molecules28135102_

Round 1

Reviewer 1 Report

In this manuscript, Gao et.al. investigated the extraction methods of polysaccharides from Phellinus linteus and used different methods to characterize the extracts. However, some of the characterizations are not conclusive, and I listed some questions here.

Section 2.1, what is PMP?

Section 2.3, how to generate the Phellinus linteus mycelia powder? Also, in section 2.6, it’s very confusing that the PL and PIP powder extraction is described again. Then for the following experiments, are they extracts from 2.3 or 2.6? (From my understanding, the authors are trying to sat PIP and PEP, but wrote in mistake?)

In section 2.6.3, there’s no information about the Coomassie brilliant blue concentration or amount used. Also, I would call the reading as “absorbance” instead of “OD”.

In section 2.6.5, I think the author is trying to say “pipette” instead of “gun”?

I think the model fitting part is interesting. But how big is the difference between yield 13% and 10%? Are there some more extreme conditions and really low/high yield to show the application of the model?

Figure 8. Same retention time on the HPLC doesn’t mean it’s the same compound. 

I don’t quite understand why the authors need to do the SEM experiments. After extraction, the morphology can be greatly changed. If you really want to do a SEM experiment, why not use the growth medium to see the mycelia and the mushroom itself?

There’s no control for the section 3.7 antimicrobial experiments so I would not trust the results. 

There are many small typos in this paper, missing some Italic fonts, and the English writing needs some improvement.

Author Response

Dear editor:and Reviewers

I am the writer of the article "Study on extraction, Physicochemical properties, and bacterio-static activity of Polysaccharides from Phellinus linteus" (ID:2435579). I am grateful to the editors and reviewers for their suggestions for the revision of this article, and according to the suggestions careful modifications have been made. We will now report to you the status of the revision of this article.

For language, we have invited professionals to make modifications.

1. Section 2.1, what is PMP?

The problem has been corrected.

“Anhydrous ethanol, glucose, phenol, sulfuric acid, 1-Phenyl-3-methyl-5-pyrazolone (PMP), and analytical purity were purchased from Sinopharm Chemical Reagent Co(Shanghai, China).”

2. Section 2.3, how to generate the Phellinus linteus mycelia powder? Also, in section 2.6, it’s very confusing that the PL and PIP powder extraction is described again. Then for the following experiments, are they extracts from 2.3 or 2.6? (From my understanding, the authors are trying to sat PIP and PEP, but wrote in mistake?)

The problem has been corrected.

2.3. The extraction process of PLP

PL fungus was inoculated into liquid medium (MgSO4 0.5g/L, KH2PO4 0.5g/L, NaCl 1.0g/L, sucrose: beef extract 40:10(g/g) with water to 100mL) at 26℃, 150r/min incubator for 9 days.

The fermentation broth of PL was centrifuged, the supernatant was extracted, and the concentrate was obtained by rotary evaporation, 4 times the volume of ethanol was added, placed at 4℃ overnight, centrifuged and precipitated by freeze-drying to obtain PEP powder.

The appropriate amount of PL powder was weighed, extracted by ultrasonic-assisted hot water extraction, the supernatant was rotary evaporated to obtain the concentrate, added 4 times the volume of ethanol, placed at 4℃ overnight, centrifuged and precipitated freeze-dried to obtain PIP powder.

3. In section 2.6.3, there's no information about the Coomassie brilliant blue concentration or amount used. Also, I would call the reading as “absorbance” instead of “OD”.

The problem has been corrected.

“Add 0.0, 0.1, 0.2, 0.3, 0.4, 0.5, and 0.6 mL of the solution to a stoppered test tube, add water to 1.0 mL of each, then add 2.5 mL of 0.1 mg/mL of coomassie brilliant blue G250 to the test tube and shake well, and let stand for 5 min at room temperature. The absorbance of the sample was measured at 595 nm, and the standard curve was plotted using the mass of bovine serum albumin contained (μg) as the horizontal coordinate and the corresponding absorbance as the vertical coordinate”

4. In section 2.6.5, I think the author is trying to say “pipette” instead of “gun”?

The problem has been corrected.

“Add 0.5 mL of 0.5M PMP-methanol, at which time PMP and NaOH will diffuse rapidly and mix, and blow the insoluble material at the bottom with a pipette to disperse it evenly, and then aspirate 0.2 mL of the mixture in an EP tube, and put the sample into a water bath at 70°C for 30 min.”

5. I think the model fitting part is interesting. But how big is the difference between yield 13% and 10%? Are there some more extreme conditions and really low/high yield to show the application of the model?

Yield 13% and 10% is what percent of crude polysaccharide is in 1 g of phellinus linteus powder, with a 30 mg difference between them.These are relatively reasonable conditions designed according to Design-Expert 8.0.6 software Box-Behnken and considered usable.

6. Figure 8. Same retention time on the HPLC doesn’t mean it’s the same compound.

The problem has been corrected.

Indeed, the same retention time for the sample and the standard in HPLC does not represent the same compound. However, we already know the sample to some extent, so we speculate that the sample is that compound and have modified it in the article.

“As shown in Figure 8, the monosaccharide composition of the PIP and PEP was de-termined by HPLC. We speculate that the two polysaccharides are heteropolysaccharides made up of multiple monosaccharides.”

7. I don’t quite understand why the authors need to do the SEM experiments. After extraction, the morphology can be greatly changed. If you really want to do a SEM experiment, why not use the growth medium to see the mycelia and the mushroom itself?

There are few articles on SEM observations of mycelial polysaccharides in phellinus linteus, so we wanted to examine the difference between PIP and PEP. To provide some experimental data for applying mycelial polysaccharides of phellinus linteus.

8. There’s no control for the section 3.7 antimicrobial experiments so I would not trust the results.

We calculated the inhibition rate by substituting the data from the control experiment into the equation in section 2.7.

If you have any questions about the above answers, please contact me

Yours sincerely,

Lihong Zhang

Reviewer 2 Report

The topic is interesting, but some suggestions are needed to lead the changes that would improve the article.

- It would be easier if the lines were numbered.

- the title contains the word "preparation". Please replace it with extraction or something similar.

- "Korea, Chin, and Japan" is China?

- it is not necessary to present the " standard curves". It is prepared during the experiment but it isn't necessary to add in the article.

- "sterile centrifuge tube"  I recommend capital letter

- of course, it was determined the standard curve but is not necessary to present it.

In my opinion, the article could be accepted after some revisions.

English needs minor corrections.

Author Response

Dear editor and Reviewers:

I am the writer of the article "Study on extraction, Physicochemical properties, and bacterio-static activity of Polysaccharides from Phellinus linteus" (ID:2435579). I am grateful to the editors and reviewers for their suggestions for the revision of this article, and according to the suggestions careful modifications have been made. We will now report to you the status of the revision of this article.
For language, we have invited professionals to make modifications.

1. It would be easier if the lines were numbered.
The problem has been corrected.

2. the title contains the word "preparation". Please replace it with extraction or something similar.
The problem has been corrected.
“Study on extraction, Physicochemical properties, and bacterio-static activity of Polysaccharides from Phellinus linteus”

3. "Korea, Chin, and Japan" is China?
The problem has been corrected.
“As one of many medicinal mushrooms, it is widely used in East Asia, especially in Korea, China, and Japan.”

4. It is not necessary to present the " standard curves". It is prepared during the experiment but it isn't necessary to add in the article.
The problem has been corrected.

5. "sterile centrifuge tube"  I recommend capital letter
The problem has been corrected.
“The sterile centrifuge tube was boiled at 100℃ for 20 min and sterilized by UV irradiation for 30 min, 0.5 mL of each polysaccharide solution was taken.”

If you have any questions about the above answers, please contact me
Yours sincerely,

Lihong Zhang

Reviewer 3 Report

Dear Authors,

I regret to say that I do not recommend your article for publication in Molecules. The work is valuable, the authors put a lot of time into the preparation of the research and interpretation of the results, but the point "Materials and methods" needs complete improvement. As for me, it is written very chaotically, and I suspect that scientists who would like to recreate some of the research described would not be able to do so. Point “Results and Discussion”, as the name suggests, should refer to the comparison of the research results obtained by the authors with the research of other scientists - I miss a discussion here. In addition, attention should be paid to the English language, punctuation, and expansion of abbreviations used in the text of the publication

Author Response

I am the writer of the article "Study on extraction, Physicochemical properties, and bacterio-static activity of Polysaccharides from Phellinus linteus" (ID:2435579). I am grateful to the editors and reviewers for their suggestions for the revision of this article, and according to the suggestions careful modifications have been made. This article has been modified in terms of content and paragraph layout, and for the language, we have invited professionals to make alterations. 
We deeply appreciate your consideration of our manuscript, and we look forward to receiving comments from the reviewers. If you have any queries, please don’t hesitate to contact me at the address above
Yours sincerely,

Lihong Zhang

Round 2

Reviewer 1 Report

I can see that the discussion in section 3.2-3.4 has greatly improved. However, I still don’t receive an answer for my previous question on the control in Figure 10. You need both positive controls and negative controls to demonstrate your antimicrobial assay is effective. 

New questions I have:

You have sucrose in your culture medium. How do you justify the glucose and fructose in your figure 6 is not coming from the medium?

Line 50-51: what does it mean by saying it does not need to consider the time factor?

Figure 6: please list the full name of these monosaccharides somewhere in your main text

Minor issues:

Line 194: The authors should keep using “OD” for the bacterial growth optical density measurement. There’s a difference between “absorbance” and “OD” and not everything needs to change to “absorbance”.

Line 85: beef extract is mentioned in the medium component, but not mentioned in section 2.1.

Line 87: extract by what? Ethanol? (mention ethanol in line 87 first instead of line 89)

I still find some typos in the manuscript, and some sentences are not a complete sentence. I suggest the authors pay more attention to the writing. For example:

Line 30: “Phellinus” (Capitalize)

Line 43: “color” is actually correct

Line 57: not a complete sentence

Line 56: you either say “anti-virus” or “anti-viral reagent”. Same for all of the others

Line 159: “wholly” -> “completely”; add a space between “0.5M”: “0.5 M”. Same for all others.

Line: 184: italic for “E.coli” and other species

Line 398 line 401 line 403: italic font

Line 404: “more potent”

Author Response

Dear editor and Reviewers:

I am the writer of the article "Study on extraction, Physicochemical properties, and bacteriostatic activity of Polysaccharides from Phellinus linteus" (ID:2435579). I am grateful to the editors and reviewers for their suggestions for the revision of this article, and according to the suggestions careful modifications have been made. We will now report to you the status of the revision of this article.

For language, we have invited professionals to make modifications.

We hope that the revision is acceptable, and I look forward to hearing from you soon.

Yours sincerely,

Lihong Zhang

We would like to express our sincere thanks to the reviewers for their constructive and positive comments. As I have re-edited the language and structure, the overall structure of the article has changed a little and the line numbers are different from before. For your convenience, I will mark the line numbers that have changed after each change.

1. You need both positive controls and negative controls to demonstrate your antimicrobial assay is effective.

The problem has been corrected.

In the preliminary experiments, we made a control (sterile water) and an experimental group (polysaccharide solution), tested the OD600 values under the same incubation conditions and calculated the inhibition rate by substituting them into equation (ratio of control-experimental group difference to control group) (2). Because the OD values of the control were taken into the equation and the inhibition rate was 0%, they are not reflected in the graph.

We have added a related thesis, "Mutation breeding of bacteriocin-like substance-producing Lactococcus lactis NFL and the optimization of determinationmethod of the antimicrobial activity of bacteriocin-like substance" (reference 36). We hope this will be useful for your review.

Thank you very much for your review, and I hope to receive your approval.

2. You have sucrose in your culture medium. How do you justify the glucose and fructose in your figure 6 is not coming from the medium?

The problem has been corrected.

The PIP is extracted by washing the mycelium several times with deionized water and then extracting. Also because both PIP and PEP are alcohol-precipitation method during extraction, the oligosaccharides (sucrose) monosaccharides are generally in the alcohol-precipitation supernatant, so the glucose and fructose in Figure 6 would not be the sucrose inside the medium.

“The PL fermentation broth was centrifuged to obtain the supernatant, and the concentrate was obtained by rotary evaporation, 4 times the volume of ethanol was added, placed at 4℃ overnight, centrifuged and precipitated by freeze-drying to obtain PEP powder.

The appropriate amount of PL powder was weighed, extracted by ultrasonic-assisted hot water extraction, the supernatant was rotary evaporated to obtain the concentrate, added 4 times the volume of ethanol, placed at 4℃ overnight, centrifuged and precipitated freeze-dried to obtain PIP powder.” (line375-382)

3. Line 50-51: what does it mean by saying it does not need to consider the time factor?

The problem has been corrected.

“Due to the growth habit of the fungus, it is possible to reduce the preparation time of fungal polysaccharides drastically.”(line46-47)

4. Figure 6: please list the full name of these monosaccharides somewhere in your main text

The problem has been corrected.

3.1. Chemicals and Reagents.

“Monosaccharide standards, including Arabinose (Ara), Fucose (Fuc), Galactose (Gal), Galacturonic acid (GalA), Glucose (Glc), Glucuronic acid (GlcA), Mannose (Man), Rham-nose (Rha), Xylose (Xyl), the chromatographic purity were purchased from Shanghai-yuanyeBio-TechnologyCo. (Shanghai, China).” (line251-254)

5. Line 194: The authors should keep using “OD” for the bacterial growth optical density measurement. There’s a difference between “absorbance” and “OD” and not everything needs to change to “absorbance”.

The problem has been corrected.

“The sterile centrifuge tube was boiled at 100℃ for 20 min and sterilized by UV irradi-ation for 30 min, 0.5 mL of each polysaccharide solution was taken. Equal amounts of S. aureus, E. coli, and B. subtilis were added, shaken at 37℃, 200 r/min and incubated at 0, 1.0, 1.5, 2.0, 2.5, 3.0, 3.5, 4.0 h, respectively. The samples were taken at 0, 1.0, 1.5, 2.0, 2.5, 3.0, 3.5, 4.0 h, the OD value was measured at 600 nm, and three parallel groups were made in each group. A line graph was plotted with time as the horizontal coordinate and inhibition rate as the vertical coordinate for analysis.”

(line381-382)

6. Line 85: beef extract is mentioned in the medium component, but not mentioned in section 2.1.

The problem has been corrected.

“Pancreatic peptone, yeast extract powder, beef extract, biochemical reagent were purchased from Beijing Auboxing Biotechnology Co (Beijing, China).” (line259-260)

7. Line 87: extract by what? Ethanol? (mention ethanol in line 87 first instead of line 89)

The problem has been corrected. The supernatant can be obtained directly after centrifugation without extraction.

The PL fermentation broth was centrifuged to obtain the supernatant, and the concentrate was obtained by rotary evaporation, 4 times the volume of ethanol was added, placed at 4℃ overnight, centrifuged and precipitated by freeze-drying to obtain PEP powder.” (line276-279)

8. Line 30: “Phellinus” (Capitalize)

The problem has been corrected.

Phellinus linteus (PL) is a rare large fungus; PL is also known as mulberrychen, tree chicken, plum tree fungus, etc., belongs to the fungus world, stretcher fungus, umbrella fungus, rust Gekomycetes, phellinus linteus genus, mainly produced in tropical America, Africa, and East Asia.” (line28-31)

9. Line 43:“color” is actually correct

The problem has been corrected.

“So the yellow of PL refers to the color or texture.” (line40)

10. Line 57: not a complete sentence

The problem has been corrected.

“An increasing number of structurally diverse polysaccharides have been obtained from phellinus linteus cotyledons, phellinus linteus mycelium, and fermentation broth by a large number of extraction and separation methods, such as hot water extraction, ultrasonic ex-traction, enzymatic extraction, and a large number of purification methods such as ethanol precipitation and packed column chromatography.” (line54-58)

11. Line 56: you either say “anti-virus” or “anti-viral reagent”. Same for all of the others.

The problem has been corrected. We have checked the full text for relevant issues.

“These include anti-oxidation , antitumor, immunomodulatory, anti-inflammatory, anti-virus, free-radical scavenging ability, and antibacterial effects.” (line52-54)

12. Line 159: “wholly” -> “completely”; add a space between “0.5M”: “0.5 M”. Same for all others.

The problem has been corrected.

“Weigh 2 mg of the sample into the acid hydrolysis vial, add 1 mL of methanolic solution of hydrochloric acid, charge with N2, and then react for 16 h at 80℃ in a constant temperature metal bath. After the reaction, the methanol of hydrochloric acid in the sam-ple was blown dry with a nitrogen-blowing instrument. Then 1 mL of 2 M trifluoroacetic acid (TFA) was added to the reaction at 120℃ for 1 h. After the reaction, it was blown dry again. Next, monosaccharide derivatization was performed by adding 0.5 mL of 0.3 M NaOH to the acid hydrolysis vial to dissolve the dried monosaccharide sample inside completely. Add 0.5 mL of 0.5 M PMP-methanol, at which time PMP and NaOH will dif-fuse rapidly and mix, and blow the insoluble material at the bottom with a pipette to disperse it evenly, and then aspirate 0.2 mL of the mixture in an EP tube, and put the sample into a water bath at 70°C for 30 min. Samples were taken after the reaction and extracted after adding 0.1 mL of 0.3 M HCl. Add 0.7 mL of CH₂Cl2 and shake the EP tube for 90 s until complete mixing and centrifugation to completely separate the organic phase from the aqueous phase. Pump the lower organic phase clean with a flat-tipped syringe and repeat twice. The remaining aqueous phase was pumped up with a 1 mL syringe and then filtered through an organic filter membrane of 0.22 μm, and the sample was subsequently placed in HPLC for detection.

HPLC analysis method: Shimadzu HPLC system (LC-20ATvp pump and SPD-20AVD UV detector), COSMOSIL 5C18-PAQ column (4.61×250 mm), mobile phase PBS (0.1 M, pH 7.0)-acetonitrile 80.8:19.2 (v/v) at a flow rate of 1.0 mL/min, with an injection volume of 20 μL and detection at 245 nm.” (line345-365)

13. Line: 184: italic for “E.coli” and other species

The problem has been corrected.

“Under aseptic conditions, take the cryopreserved E. coli, S. aureus and B. subtilis into LB medium and activate them at 200 rpm, 37℃ for 12 h.” (line371-373)

14. Line 398 line 401 line 403: italic font

The problem has been corrected.

“As can be seen from Figure 10, the PIP gradually decreased the inhibition rate of S. aureus with the growth of time, and the PEP was extended with the shaking culture time, the trend was the same as the PIP, and the inhibition rate was gradually decreased. The strongest inhibitory effect of PIP on S. aureus, with an inhibition rate of 80% at 1.0 h. The inhibition rate of the PIP against E. coli was the highest at 1.5 h, with 75.47%. The inhibi-tion of E. coli by PEP gradually decreased with time. The two polysaccharides were slightly less effective in inhibiting B. subtilis, and the inhibition rate gradually decreased with time. The PIP had the best effect on its inhibition, with the highest inhibition rate of 67.74%. The inhibitory effect of PIP on B. subtilis was more potent than that of PEP.” (line235-243)

15.Line 404: “more potent”

The problem has been corrected.

“The inhibitory effect of PIP on B. subtilis was more potent than that of PEP.” (line242-243)

Reviewer 3 Report

Dear Authors,

I recommend publishing in the current, corrected version

Author Response

Dear editor and Reviewers:

I am the writer of the article "Study on extraction, Physicochemical properties, and bacteriostatic activity of Polysaccharides from Phellinus linteus" (ID:2435579).I am very grateful for your review and endorsement.

Yours sincerely,

Lihong Zhang